# Hemodialysis without Systemic Anticoagulation: A Randomized Controlled Trial to Evaluate Five Strategies in Patients at a High Risk of Bleeding

**DOI:** 10.3390/medsci12030038

**Published:** 2024-08-04

**Authors:** Pedro H. Franca Gois, David McIntyre, Sharad Ratanjee, Anita Pelecanos, Carla Scuderi, Chungun L. Janoschka, Kara Summers, Haibing Wu, Belinda Elford, Dwarakanathan Ranganathan, Helen G. Healy

**Affiliations:** 1Kidney Health Service, Royal Brisbane and Women’s Hospital, Herston, QLD 4029, Australia; sharad.ratanjee@health.qld.gov.au (S.R.); carla.scuderi@health.qld.gov.au (C.S.); chungun.lee@health.qld.gov.au (C.L.J.); kara.summers@health.qld.gov.au (K.S.); haibing.wu@health.qld.gov.au (H.W.); belinda.elford@health.qld.gov.au (B.E.); dwarakanathan.ranganathan@health.qld.gov.au (D.R.); helen.healy@health.qld.gov.au (H.G.H.); 2Faculty of Medicine, University of Queensland, Herston, QLD 4006, Australia; 3Nephrology Department, John Hunter Hospital, New Lambton Heights, NSW 2305, Australia; 4Nephrology Department, Townsville University Hospital, Douglas, QLD 4814, Australia; david.mcintyre@health.qld.gov.au; 5Statistics Unit, QIMR Berghofer Medical Research Institute, Herston, QLD 4006, Australia; anita.pelecanos@qimrberghofer.edu.au; 6Conjoint Kidney Research Laboratory, Chemical Pathology—Pathology Queensland, Herston, QLD 4006, Australia

**Keywords:** Hemodialysis, anticoagulation, heparin-free, intermittent saline flushes, heparin-coated dialyzer, randomized controlled trial

## Abstract

Background: There has been growing interest in exploring combined interventions to achieve a more effective heparin-free treatment approach. Aim: to evaluate combination of interventions compared to standard practice (intermittent flushes) to prevent clotting and consequently reduce premature interruptions of hemodialysis. Methods: This open-label randomized controlled trial recruited chronic hemodialysis patients with contra-indication to systemic heparinization. Participants were randomized into one of five groups to receive different strategies of heparin-free hemodialysis treatment for up to three sessions. Primary endpoint: the successful completion of hemodialysis without clotting. Secondary outcomes: the clotting of the air traps assessed by a semi-quantitative scale, online KT/V, and safety of the interventions. Results: Forty participants were recruited and randomized between May and December 2020. Participants showed similar baseline biochemistry results and coagulation profiles. The highest success rates were observed in group 3 (heparin-coated dialyzers combined with intermittent flushes) (100%) and group 5 (hemodiafiltration with online predilution combined with heparin-coated dialyzers), with 91% vs. the control (intermittent flushes) (64%). Group 2 (heparin-coated dialyzers alone) had the poorest success rate, with 38% of the sessions being prematurely terminated due to clotting. KT/V and clotting scores were similar between groups. No adverse events related to the trial interventions were observed. Conclusions: The proposed combination of interventions may have had additive effects, leading to less frequent clotting and the premature termination of an HD/HDF session. Our study supports the feasibility of conducting a larger randomized controlled trial focusing on the efficacy of combined interventions for heparin-free HD in patients with a high risk of bleeding.

## 1. Introduction

Hemodialysis (HD) and, more recently, hemodiafiltration (HDF) are life-sustaining therapies for individuals with kidney failure. These treatments involve the passage of blood through an extracorporeal circuit. This contact of the blood with a non-physiological environment leads to clotting, which reduces the circuit lifetime and increases patients’ blood loss and material consumption, ultimately resulting in a lower efficiency and higher cost of the treatment [1].

Heparin, either unfractionated (UFH) or low molecular weight (LMWH), is commonly used to prevent clot formation and the interruption of an HD session [2,3]. An efficient and safe HD session requires a subtle balance between under- and over-heparinization to prevent clotting and bleeding, respectively [1]. The tailoring of the anticoagulant prescription incorporates estimates of the risk of bleeding for each patient [1].

Systemic anticoagulation should rationally be avoided in patients at a high risk of bleeding [1,4]. Several methods have been proposed to avoid the clotting of the extracorporeal circuit in scenarios of this clinical risk [2]. The European Best Practice Guidelines on HD recommend intermittent 0.9% saline flushes as the method of choice for patients at a high risk of bleeding, although the evidence for this practice is somewhat outdated and not based on randomized trials [4].

Since 2002, other options for heparin-free HD have been proposed, including HDF with online predilution and limiting heparin to the extracorporeal circuits. Several studies investigated the use of heparin-coated dialyzers in patients at a high risk of bleeding, with inconsistent clotting rates of up to 49.6% reported [2,5,6,7,8].

The safety of patients exposed to a procedure with a high risk of clotting is a compelling rationale to study strategies such as the combination of standard interventions to prevent premature interruptions of an HD treatment session. There has been growing interest in exploring combined interventions to achieve a more effective heparin-free treatment approach [9,10,11,12]. The recently published literature has primarily focused on the combination of heparin-coated dialyzers with citrate-enriched dialysate [10,11,12]. Nevertheless, a head-to-head comparison of combined interventions (i.e., heparin-coated dialyzer + intermittent 0.9% saline flushes and heparin-coated dialyzer + HDF with online predilution) with any other single interventions (i.e., intermittent 0.9% saline flushes or HDF with online predilution alone) has not been carried out to date. The aim of this study was to conduct a study to evaluate the performance of combinations of interventions to prevent the clotting of dialyzer/lines compared to single interventions in patients with a high risk of bleeding who require heparin-free HD.

## 2. Materials and Methods

This clinical study was an open-label randomized controlled trial, conducted in a single tertiary hospital and health service (comprising the Royal Brisbane and Women’s Hospital HD unit and the Redcliffe Satellite HD unit) between May and December 2020. Consecutive chronic HD patients with contra-indication to systemic heparinization were screened by the investigators. Signed informed consent was obtained from eligible participants. Individuals were randomly allocated using a computer-generated randomization (http://www.randomization.com) (accessed on 12 February 2018) into five groups to receive different strategies of heparin-free HD treatment (Figure 1). Participants were enrolled and allocated to one of the study groups by a clinical research nurse, who was not involved in the care of the patient during the HD/HDF session.

### 2.1. Study Groups

Group 1 (control) involving intermittent 0.9% saline flushes (100 mL per flush every 30 min during treatment) was considered the control group, as per the European Best Practice Guidelines in HD.

### 2.2. Interventions

Group 2 comprised participants allocated to receive HD with a heparin-coated dialyzer (Evodial^®^, Gambro, Sydney, Australia). Group 3 participants received a combination of a heparin-coated dialyzer (Evodial^®^) and intermittent 0.9% saline flushes. Group 4 received HDF with online predilution and Group 5 received a combination of HDF with online predilution and a heparin-coated dialyzer (Evodial^®^).

Both HD and HDF were performed with ultrapure bicarbonate-based dialysis fluids. In HDF, the convection volume was driven automatically using the “auto sub” system of the Fresenius 5008 machine. Evodial^®^ (2.2) dialyzers were purchased from Gambro^®^ (Sydney, Australia) and Elisio 21H^®^ dialyzers from Nipro^®^ (St Leonards, NSW, Australia). All equipment and devices used in this research are approved for use by the Therapeutic Goods Administration, Australian Federal Government.

Participants remained in their allocated group for up to a maximum of three consecutive heparin-free HD/HDF sessions, without any switch allowed between arms. Any fluid infused as part of the anticoagulant strategy was added to the patient’s ultrafiltration goal. All other HD/HDF parameters, such as pump speed, treatment duration, ultrafiltration, and access cannulation, were determined by the participants’ treating nephrologists.

### 2.3. Primary Outcomes

The primary outcomes were the successful completion of HD or HDF without significant problems with blood clotting. The primary endpoint was assessed by the assisting HD nurse. HD/HDF treatments were considered successful when the following occurred:No complete occlusion of air traps or dialyzer rendering HD impossible (grade 4 according to a semi-quantitative scale previously published) [6].No additional 0.9% saline flushes to prevent clotting.No exchange of dialyzer or bloodlines because of clotting.No premature termination (early rinse back) because of clotting.

### 2.4. Secondary Outcomes

The follow-up of clotting in the air traps assessed using a 4-point semi-quantitative scale was a secondary outcome of study. Grade 1 was no detectable clotting, grade 2 was minimal clot formation—fibrinous ring, grade 3 was clot formation (up to 5 cm) but with HD still possible and grade 4 was the complete occlusion of air traps or a dialyzer rendering HD termination. Other secondary outcomes were HD adequacy (online KT/V) and safety of the interventions, i.e., the documentation of adverse events during an HD/HDF session.

Clot grading was performed hourly by two independent observers. These were either nurses (the participant’s primary nurse and a second nurse not in charge of the patient) or the primary nurse and a study investigator.

In case of discordance between the two observers or in the case of premature session termination (grade 4), the final decision was adjudicated by a third authorized and trained person (i.e., one of the chief or associate investigators). At each site and prior to the enrolment of patients, all those involved in the study were trained to use the clotting scale.

Nursing staff received training provided by Gambro^®^ (manufacturer of Evodial^®^) in the use of the dialyzer, including the recommended priming procedures. Gambro^®^ provided this training without any financial compensation or participation in the study. The primary HD nurse was accountable for delivering the anticoagulant strategy according to the allocated group.

### 2.5. Study Population

Inclusion criteria were the following: age ≥18 years; individuals with kidney failure on maintenance HD or HDF for more than 3 months; individuals receiving in-center HD; a high risk of bleeding, as assessed by the attending renal physician; medical decision by the attending renal physician to perform heparin-free HD; and either native or graft arteriovenous fistula with a blood flow of at least 250 mL/min or a tunneled central venous catheter locked by heparin. In this instance, heparin is removed from the lumen and the catheter is flushed with 0.9% saline prior to starting the anticoagulant strategy.

Exclusion criteria were as follows: dialysis in the intensive care unit; acute kidney injury; vascular access via a single needle; known heparin contra-indication (e.g., heparin-induced thrombocytopenia type II); the requirement of blood products during HD treatment; individuals receiving oral anticoagulants (including anti-vitamin K); individuals receiving a combination of anti-platelet agents; patients receiving either UFH or LMWH to treat deep vein thrombosis; patients receiving either UFH or LMWH to prevent deep vein thrombosis; laboratory markers of liver dysfunction (ALT and AST > 2 times the upper limit of the reference range); known coagulopathy or hemostasis disorder (pathological value of prothrombin time and/or activated partial thromboplastin clotting time and/or platelets <50,000/uL); and individuals with a malignant diagnosis. Individuals were considered for inclusion if oral anticoagulation was ceased for >3 days and they had a documented normal international normalized ratio, dual antiplatelet therapy was stopped for >5 days, and LMWH was ceased for >24 h (or >12 h for UFH).

Data including participants’ demographics, laboratory, and medical information (i.e., age, gender, HD vintage, primary renal disease, main comorbidities, concomitant medications including erythropoietin dosing, and indication for heparin-free HD) were extracted from the participants’ clinical record. Data were then entered on purpose-built templated data collection sheets using REDCap^®^ electronic data capture tools hosted at Metro North Hospital and Health Service. Participants were de-identified prior to data extraction and analysis.

### 2.6. Statistical Methods

Based on previous studies [2,6,8], the rates of the primary outcomes were 67% with intermittent 0.9% saline flushes, 37% with heparin-coated circuits, and 60% with pre-dilution HDF. Using a two-tailed z-test of proportions between groups with 80% power and a 5% level of significance, we estimate a sample size of 40 patients in each arm to detect a significant difference of 30% between intermittent 0.9% saline flushes and heparin-coated circuits, whereas a sample size of 73 patients would detect a difference of 7% between intermittent 0.9% saline flushes and pre-dilution HDF.

Data were analyzed according to an intention-to-treat analysis. Variables were summarized using medians (interquartile range) for continuous variables and numbers (percent) for categorical variables. As some patients were included in the trial more than once (more than one occasion of service), the analysis of outcomes was split on a per-patient and a “per occasion”-of-service basis. For the per-patient analysis, trial groups were compared using Fisher’s exact tests for categorical outcomes and Kruskal–Wallis test for continuous outcomes. Statistical significance was indicated at a *p*-value less than 0.05. For the “per occasion” of service analysis, outcomes were described using summary statistics. STATA 15 was used for the analysis.

## 3. Results

There were 137 patients assessed for eligibility, whereby 97 did not meet the inclusion criteria. A total of 40 patients were recruited and randomized, with a total of 58 sessions across all groups performed. Figure 2 presents the study flow diagram, including dropout rates and number of participants who remained in their allocated group for up to three consecutive heparin-free HD/HDF sessions. Relevant baseline characteristics of the included patients are presented in Table 1. The main indications for heparin-free HD were pre- and post-operative setting, active bleeding, and recent tunneled catheter insertion (Table 1). The recruitment process identified and enrolled individuals meeting the specific inclusion criteria, suggesting that the trial’s recruitment strategies were effective in identifying the target population within the given timeline. Additionally, our trial achieved a degree of participant stability and retention throughout the trial period.

Group 1 appeared to have a greater proportion of females compared to the other groups. Although group 2 had a large proportion of patients dialyzed via tunneled catheters compared to the other groups, we did not observe large differences in the average pump speed of the first HD/HDF session. Participants’ baseline characteristics as well as concomitant medications were otherwise similar across all groups. Antiplatelet agents were chiefly aspirin, and there were no large differences between groups. Only one patient (in group 4) was taking clopidogrel at recruitment. None of the patients included in this study were on warfarin, unfractionated heparin, LMWH or direct-acting oral anticoagulants. Several factors that could affect coagulation, such as hemoglobin, platelet count, aPTT and INR, were similar across all groups.

The rate of the primary outcome of the successful completion of HD or HDF without significant problems with blood clotting was highest in group 3 (heparin-coated dialyzers combined with IS) (100%) and group 5 (HDF combined with heparin-coated dialyzers) (83%) compared to 71% in the control group (IS alone). Group 2 (heparin-coated dialyzers alone) had the poorest success rate, with the treatment of 50% of patients being prematurely terminated due to clotting. We found a 71% success rate in group 4 (HDF with online predilution). These rates were not statistically significantly different (Table 2). Interruptions of the HD/HDF session due to clotting occurred more frequently after the third hour. KT/V and clotting scores were similar between groups (Table 3). Success rates in the first session (Table 2) were similar to the rates of all HD/HDF sessions combined (Table 4). No adverse events related to the trial interventions were observed.

## 4. Discussion

Heparin-free HD is considered in a variety of clinical settings such as recent cranial trauma, active gastrointestinal bleeding, hypertensive urgencies, pre- and post-operative states, and following biopsies and the insertion of central lines [1,4]. Despite the relevance of this topic, the available evidence is limited to only a few clinical trials comparing alternative interventions to systemic heparinization in HD. Intermittent 0.9% saline flushes have been widely used as it is deemed as a safe technique to reduce clotting. However, it requires one-to-one nursing, increasing the cost of the HD treatment and making it a poor choice for medium- and long-term outpatient use. Other options for heparin-free HD are HDF with online predilution, the use of dialyzers coated with heparin, regional anticoagulation with citrate and a citrate-enriched dialysate. Several studies have investigated the use of heparin-coated dialyzers in patients at a high risk of bleeding with contrasting results [2,5,6,7].

Laville et al. reported a multicenter, international, randomized trial comparing a heparin-coated dialyzer (Evodial^®^, Gambro, Lund, Sweden) with the standard of care, which consisted of either 0.9% saline flushes or HDF with online predilution [6]. The investigators did not show the superiority of the heparin-coated dialyzers over the standard practice. However, the overall failure rates were surprisingly high in all the studied groups, ranging from 31.5 to 49.6% [6]. We found similar numbers in our study as up to a half of the treatments using single strategies (intermittent 0.9% saline flushes, Evodial^®^ or HDF with online predilution) resulted in the premature termination of the session subsequently requiring an exchange of the whole HD/HDF circuit for the continuation of treatment.

The rationale for using predilution HDF as an alternative to systemic anticoagulation in HD is based on the theory that infused dialysate would continuously rinse the dialyzer, thereby preventing the adherence of cells and proteins implicated in blood clotting [13,14]. However, recent studies have suggested that predilution HDF may activate the coagulation process, as indicated by elevated levels of D-dimer and thrombin–antithrombin complexes [13,15]. In our trial, we introduced a predilution HDF intervention group to assess the applicability and external validity of earlier research conducted in European settings. Krummel et al. conducted a randomized trial comparing conventional HD with predilution HDF in patients with a high risk of bleeding admitted to a nephrology intensive care unit [13]. They found over 20% of premature clotting associated with predilution HDF [13], consistent with our findings in group 4.

Regional citrate anticoagulation has demonstrated significant efficacy, particularly in the intensive care unit setting [16,17]. However, it requires the monitoring of electrolyte disturbances as well as the administration of both citrate and calcium infusions, leading to increased complexity in monitoring for maintenance patients with HD [18]. Alternative approaches, such as a citrate-enriched dialysate, have been used in combination with heparin-coated membranes for heparin-free HD treatments [9,11,12]. Results from two randomized controlled cross-over trials conducted by the same renal division have been published after the completion of our study [9,11]. The EvoCit study compared the combination of a citrate-enriched dialysate + Evodial^®^ with systemic anticoagulation + Evodial^®^, while the SAFE study compared the use of a citrate-enriched dialysate + asymmetrical triacetate (ATA) dialyzer to predilution HDF + ATA [9,11]. Both studies achieved success rates of >90% when a citrate-enriched dialysate was combined with less thrombogenic dialyzers [9,11], supporting our finding from group 3. Furthermore, the SAFE study identified success rates of 83.3% with predilution HDF + ATA [9], mirroring our observations in group 5. Unlike the EvoCit and SAFE studies, which enrolled patients on maintenance HD without an elevated risk of bleeding, our study specifically focused on individuals with a high bleeding risk necessitating heparin-free HD. While our approach reflected the real-world clinical practice, it did not favor a rapid recruitment phase.

Given the high prevalence of heparin-free HD treatments prescribed in our center, this research group decided to undergo a 12-month recruitment phase which led to recruitment of at least 150 participants, providing data for at least 200 HD/HDF sessions, considering how frequent our study participants remained in the trial for subsequent sessions. Nevertheless, this study was profoundly impacted by the COVID19 pandemic, particularly resulting in cancelations of elective and semi-elective surgeries and procedures. The budget allocated to this study did not allow an extension of the recruitment period, resulting in the study being terminated and not attaining the pre-planned study participant numbers. While our study lacked the power to detect statistically significant differences, our goal is to report trial group statistics for sample-size planning in a future larger-scale trial, aiming for more robust conclusions.

Some studies have shown that men may have a slightly higher overall risk of thrombosis than women, but women have additional risks due to birth control and postmenopausal hormone therapy [19]. Although group 1 had a greater proportion of females compared to the other groups, none of these patients were on estrogen-based hormone therapy during their participation in the study; hence, it was an unlikely source of bias. Furthermore, despite central randomization, group 2 had a proportion of patients dialyzing via a tunneled catheter. Sahota et al. have shown that a lower access blood flow, rather than the type of access used, was associated with higher clotting rates of the HD circuit [20]. Reassuringly, this was not a potential source of bias, and we found no difference in pump speed not only across all groups but also when we analyzed pump speed in different types of HD access. However, given the small sample size, it is difficult to tease out these effects.

Acute reactions to dialyzers in HD are uncommon and can be graded as mild, moderate, or severe [21,22]. Symptoms may include shortness of breath, nausea, and low blood pressure. Only one case has been reported in the literature of a non-fatal allergic reaction to Evodial^®^ [23]. Extra care must be taken when treating patients for initial treatments with a specific dialyzer. We have monitored our trial participants hourly for the occurrence of possible hypersensitivity symptoms. We did not record any side effects related to sensitivity reactions to any of the dialyzers used in this study.

A single episode of clotting leading to an exchange of the extracorporeal circuits almost doubles the clinical waste generated during HD. This is particularly relevant considering the vast quantities of waste generated in HD provision, i.e., a single HD session produces about 2.5 kg of solid clinical waste, of which 38% is plastic [24]. Moreover, although the combination of strategies did not result in better HD adequacy evaluated by the online KT/V, clotting and eventually the exchanging of the dialyzer and lines, it may lead to higher HD downtime, blood losses, and more clinical waste generation.

This study has several limitations. Firstly, given its early termination, our study was underpowered to detect significant differences between the study groups. If our study had been designed to include patients without a high risk of bleeding, it might have boosted recruitment rates. However, this approach would not have directly targeted the specific population we intended to examine. Moreover, our research team recognized the ethical dilemma of subjecting patients to an elevated risk of circuit clotting without a clear indication to perform heparin-free HD. Secondly, our study design was somewhat complex and less pragmatic. Opting for a five-arm study was driven by the need of assessing the real-world feasibility and validity of these interventions within the specific milieu of a tertiary healthcare facility in Australia. Thirdly, gender and vascular access type was unbalanced across the study groups despite randomization. Fourthly, during the recruitment phase, a citrate-enriched dialysate was not available in our hospital and health service. Only 15% of all dialysis treatments in Australia utilize a citrate-enriched dialysate, with an even lower adoption rate of around 3% in the state of Queensland. All our patients were dialyzed with the available bicarbonate-based dialysate, which may explain the higher clotting rates observed when single interventions were employed.

To our knowledge, this is the first study evaluating the combination of saline flushes and predilution HDF with heparin-coated dialyzers to prevent the clotting of the extracorporeal circuit in chronic HD patients with a high risk of bleeding, hence providing at least two additional alternative strategies to prevent clotting in heparin-free HD sessions. The proposed combination of interventions, such as heparin-coated dialyzers combined with intermittent 0.9% saline flushes and predilution HDF combined with heparin-coated dialyzers, may have had additive effects, leading to less frequent clotting and a premature termination of an HD/HDF session. Our study supports the feasibility of conducting a larger randomized controlled trial in a more pragmatic approach, focusing on the efficacy of combined interventions for heparin-free HD in patients with a high risk of bleeding.

## Figures and Tables

**Figure 1 medsci-12-00038-f001:**
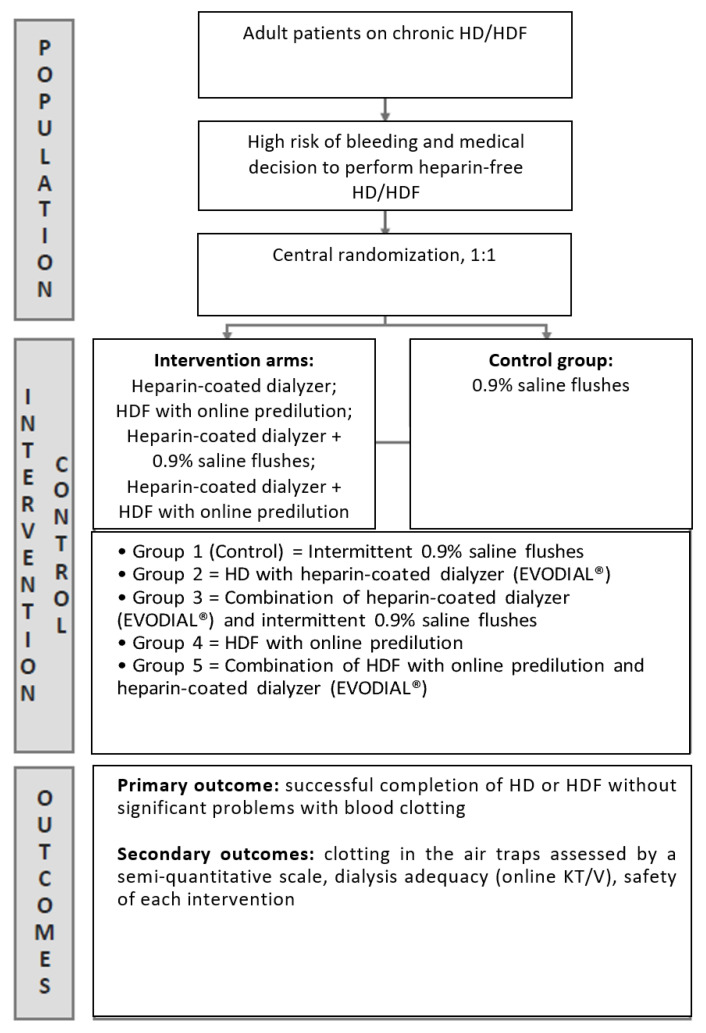
Study schema.

**Figure 2 medsci-12-00038-f002:**
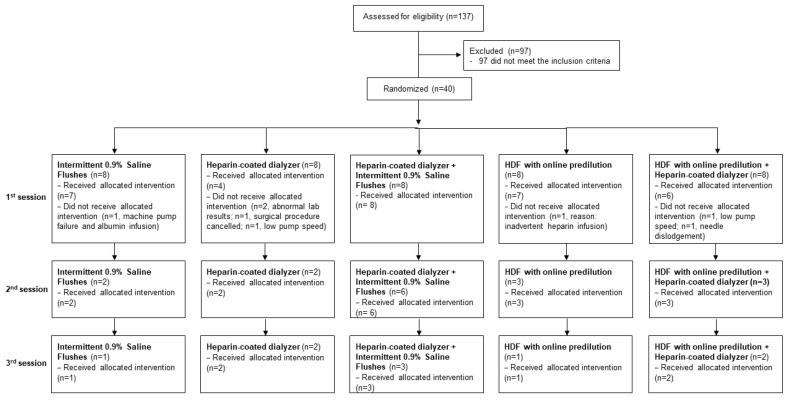
CONSORT 2010 flow diagram of the study.

**Table 1 medsci-12-00038-t001:** Patient characteristics.

	Group 1 Intermittent 0.9% Saline Flushes	Group 2 Evodial^®^	Group 3 Evodial^®^ + Intermittent 0.9% Saline Flushes	Group 4 HDF	Group 5 HDF + Evodial^®^
	*n* = 7	*n* = 4	*n* = 8	*n* = 7	*n* = 6
Age	62 (36–67)	72 (62–78)	69 (51–77)	57 (38–59)	58 (46–79)
Male gender	3 (43%)	3 (75%)	7 (88%)	4 (57%)	5 (83%)
Dialysis vintage (years), (*n* = 30)					
2005–2014	2 (29%)	2 (50%)	5 (63%)	1 17%)	0 (0%)
2015–2020	5 (71%)	2 (50%)	3 (38%)	5 (83%)	5 (100%)
Primary renal disease					
Glomerulonephritis	1 (14%)	0 (0%)	1 (13%)	1 (14%)	0 (0%)
Diabetic nephropathy	4 (57%)	2 (50%)	2 (25%)	1 (14%)	2 (33%)
Hypertension	0 (0%)	1 (25%)	1 (13%)	1 (14%)	0 (0%)
Polycystic kidney disease	1 (14%)	0 (0%)	1 (13%)	0 (0%	0 (0%)
Unknown	0 (0%)	1 (25%)	0 (0%)	1 (14%)	1 (17%)
Other *	1 (14%)	0 (0%)	3 (38%)	3 (43%)	3 (50%)
Main comorbidities					
Hypertension	4 (57%)	4 (100%)	4 (50%)	5 (71%)	5 (83%)
Diabetes	4 (57%)	3 (75%)	2 (25%)	2 (29%)	4 (67%)
Ischemic heart disease	2 (29%)	2 (50%)	3 (38%)	3 (43%)	1 (17%)
Peripheral arterial disease	0 (0%)	2 (50%)	2 (25%)	1 (14%)	3 (50%)
Dyslipidemia	1 (14%)	1 (25%)	3 (38%)	3 (43%)	2 (33%)
Previous DVT and/or PE	0 (0%)	1 (25%)	2 (25%)	1 (14%)	1 (17%)
Concomitant medications					
Antacid preparations *	4 (57%)	2 (50%)	7 (88%)	5 (71%)	3 (50%)
Oral hypoglycemic agents	3 (43%)	2 (50%)	2 (25%)	1 (14%)	3 (50%)
IV iron	4 (57%)	3 (75%)	6 (75%)	5 (71%)	6 (100%)
Calcium carbonate	3 (43%)	3 (75%)	5 (63%)	4 (57%)	6 (100%)
Diuretics	2 (29%)	1 (25%)	0 (0%)	1 (14%)	3 (50%)
Beta-blockers	3 (43%)	1 (25%)	3 (38%)	5 (71%)	3 (50%)
Renin–angiotensin system antagonists	2 (29%)	0 (0%)	2 (25%)	2 (29%)	1 (17%)
Erythropoietin agent	6 (86%)	0 (0%)	4 (50%)	4 (57%)	6 (100%)
Acetylsalicylic acid	3 (43%)	3 (75%)	7 (88%)	2 (29%)	6 (100%)
Pathology results prior to the heparin-free session					
Hemoglobin (g/L)	102 (83–114)	126 (102–132)	95.5 (93–100)	103 (97–110)	103 (93–108)
Haematocrit (L/L)	0.31 (0.22–0.38)	0.37 (0.28–0.41)	0.29 (0.21–0.36)	0.30 (0.26–0.35)	0.32 (0.25–0.40)
aPTT * (seconds), (*n* = 29)	29 (28–34)	35 (30–42)	28.5 (26.5–34.5)	33 (29–40)	29 (29–30)
INR *, (*n* = 28)	1 (1–1.2)	1 (1–1.1)	1.05 (1–1.1)	1 (0.9–1.1)	1 (0.9–1.1)
Creatinine (µmol/L)	419 (307–585)	854 (667–947)	550 (505–680)	661 (424–886)	752 (542–824)
Urea (mmol/L)	13.2 (8.6–18.6)	28.2 (19.5–39.9)	16.1 (13.8–19.3)	16.5 (9.5–25)	18.5 (14.4–25.2)
Albumin (g/L)	32 (24–36)	30 (25–35)	30 (28.5–33.5)	34 (28–35)	29.5 (29–36)
Corrected calcium (mmol/L)	2.41 (2.3–2.47)	2.22 (1.96–2.53)	2.44 (2.32–2.52)	2.37 (2.33–2.5)	2.45 (2.31–2.59)
Phosphate (mmol/L)	1.62 (0.66–1.74)	2.11 (1.72–2.7)	1.45 (1.24–1.96)	2.36 (1.29–2.95)	1.57 (1.28–2.32)
CRP (mg/L), (*n* = 18)	117 (22–171)	44.7 (6.4–83)	21 (5.9–28)	13 (4–43)	35 (14–56)
Platelet count (×10^9^/L)	198 (165–285)	275.5 (221–309)	258 (234–290)	210 (186–257)	181 (143–276)
Access type					
AV fistula	4 (57%)	0 (0%)	6 (75%)	3 (43%)	3 (50%)
Tunneled catheter	3 (43%)	4 (100%)	2 (25%)	4 (57%)	3 (50%)
Average pump speed—first session (mL/min)	300 (300–350)	268 (250–305)	298 (250–350)	294 (250–350)	303 (270–350)
Indications for heparin-free hemodialysis					
Active bleeding	0 (0%)	0 (0%)	3 (38%)	1 (14%)	1 (17%)
Bleeding wound	2 (29%)	0 (0%)	0 (0%)	0 (0%)	0 (0%)
Pre- and post-operative	4 (57%)	4 (100%)	3 (38%)	3 (43%)	3 (50%)
Recent biopsy	0 (0%)	0 (0%)	1 (13%)	0 (0%)	0 (0%)
Tunneled catheter insertion	1 (14%)	0 (0%)	0 (0%)	2 (29%)	2 (33%)
Recent stroke	0 (0%)	0 (0%)	1 (13%)	0 (0%)	0 (0%)
Hypertensive crisis	0 (0%)	0 (0%)	0 (0%)	1 (14%)	0 (0%)

* Other causes of primary kidney disease include cystinosis, reflux nephropathy, obstructive uropathy, renovascular disease, systemic sclerosis, Alport syndrome and acute kidney injury. Antacid preparations include proton pump inhibitors and histamine H2-receptor antagonists. aPTT: activated partial thromboplastin clotting time; INR: international normalized ratio; AV: arterio-venous.

**Table 2 medsci-12-00038-t002:** Primary outcomes (first hemodialysis session).

	Group 1 Intermittent 0.9% Saline Flushes	Group 2 Evodial^®^	Group 3 Evodial^®^ + Intermittent 0.9% Saline Flushes	Group 4 HDF	Group 5 HDF + Evodial^®^	*p*-Value
	*n* = 7	*n* = 4	*n* = 8	*n* = 7	*n* = 6	
Successful HD/HDF *	5 (71%)	2 (50%)	8 (100%)	5 (71%)	5 (83%)	0.30
Interrupted session	2 (29%)	2 (50%)	0 (0%)	2 (29%)	1 (17%)	
Exchange of dialyzer due to clotting	1 (50%)	0 (0%)	0 (0%)	2 (100%)	1 (100%)	
Early rinse back due to clotting	1 (50%)	0 (0%)	0 (0%)	0 (0%)	0 (0%)	
Additional flushes to prevent clotting	0 (0%)	0 (0%)	0 (0%)	0 (0%)	0 (0%)	
Complete occlusion of air traps	0 (0%)	2 (100%)	0 (0%)	0 (0%)	0 (0%)	
Timing of HD/HDF interruption, (*n* = 7)						
Hour 3	1 (50%)	1 (50%)	0 (0%)	1 (50%)	1 (100%)	1.00
Hour 4	1 (50%)	0 (0%)	0 (0%)	1 (50%)	0 (0%)	
Hour 5	0 (0%)	1 (50%)	0 (0%)	0 (0%)	0 (0%)	

* Successful completion of hemodialysis (HD) or hemodiafiltration (HDF) without significant problems with blood clotting.

**Table 3 medsci-12-00038-t003:** Secondary outcomes (first hemodialysis session).

		Group 1 Intermittent 0.9% Saline Flushes	Group 2 Evodial^®^	Group 3 Evodial^®^ + Intermittent 0.9% Saline Flushes	Group 4 HDF	Group 5 HDF + Evodial^®^	*p*-Value
		*n* = 7	*n* = 4	*n* = 8	*n* = 7	*n* = 6	
Online KT/V, (*n* = 16)		1.47 (1.4–1.53)	1.3 (1–1.3)	1.45 (1.18–1.5)	1.38 (1.12–1.63)	1.08 (0.84–1.5)	0.51
Clotting score (grades) *							
Hour 1, (*n* = 30)	1	6 (100%)	2 (67%)	7 (88%)	6 (86%)	6 (100%)	0.65
	2	0 (0%)	1 (33%)	1 (13%)	1 (14%)	0 (0%)	
Hour 2, (*n* = 30)	1	3 (50%)	1 (33%)	6 (75%)	6 (86%)	4 (67%)	0.50
	2	3 (50%)	2 (67%)	2 (25%)	1 (14%)	2 (33%)	
Hour 3, (*n* = 29)	1–2	2 (40%)	3 (100%)	6 (75%)	6 (86%)	5 (83%)	0.43
	3	3 (60%)	0 (0%)	2 (25%)	1 (14%)	1 (17%)	
Hour 4, (*n* = 25)	1–2	2 (50%)	2 (67%)	5 (63%)	5 (83%)	4 (100%)	0.57
	3–4	2 (50%)	1 (33%)	3 (38%)	1 (17%)	0 (0%)	

* Grade 1: no detectable clotting. Grade 2: minimal clot formation (fibrinous ring). Grade 3: clot formation (up to 5 cm) but hemodialysis still possible. Grade 4: complete occlusion of air traps or dialyzer rendering hemodialysis termination.

**Table 4 medsci-12-00038-t004:** Summary of outcomes on a “per occasion” basis across trial groups.

		Group 1 Intermittent 0.9% Saline Flushes	Group 2 Evodial^®^	Group 3 Evodial^®^ + Intermittent 0.9% Saline Flushes	Group 4 HDF	Group 5 HDF + Evodial^®^
		*n* = 11	*n* = 8	*n* = 17	*n* = 11	*n* = 11
Successful HD/HDF *		7 (64%)	3 (38%)	17 (100%)	6 (55%)	10 (91%)
Interrupted sessions		3 (30%)	5 (63%)	0 (0%)	5 (45%)	1 (9%)
Exchange of dialyzer due to clotting		2 (67%)	1 (20%)	0 (0%)	3 (60%)	1 (100%)
Early rinse back due to clotting		1 (33%)	2 (40%)	0 (0%)	1 (20%)	0 (0%)
Additional flushes to prevent clotting		0 (0%)	0 (0%)	0 (0%)	1 (20%)	0 (0%)
Complete occlusion of air traps		0 (0%)	2 (40%)	0 (0%)	0 (0%)	0 (0%)
Timing of HD/HDF interruption, (*n* = 15)						
Hour 1		0 (0%)	1 (20%)	0 (0%)	1 (20%)	0 (0%)
Hour 2		0 (0%)	0 (0%)	0 (0%)	1 (20%)	0 (0%)
Hour 3		3 (75%)	1 (20%)	0 (0%)	1 (20%)	1 (100%)
Hour 4		1 (25%)	1 (20%)	0 (0%)	2 (40%)	0 (0%)
Hour 5		0 (0%)	2 (40%)	0 (0%)	0 (0%)	0 (0%)
Clotting score (grades) ⁑						
Hour 1, (*n* = 54)	1	10 (100%)	5 (83%)	16 (94%)	7 (70%)	11 (100%)
	2	0 (0%)	1 (17%)	1 (6%)	3 (30%)	0 (0%)
Hour 2, (*n* = 54)	1	10 (100%)	6 (100%)	17 (100%)	9 (90%)	11 (100%)
	2	0 (0%)	0 (0%)	0 (0%)	1 (10%)	0 (0%)
Hour 3, (*n* = 52)	1–2	5 (63%)	5 (83%)	15 (88%)	8 (80%)	10 (91%)
	3	3 (38%)	1 (17%)	2 (12%)	2 (20%)	1 (9%)
Hour 4, (*n* = 45)	1–2	2 (33%)	3 (50%)	13 (76%)	7 (88%)	8 (100%)
	3–4	4 (67%)	3 (50%)	4 (24%)	1 (13%)	0 (0%)
Online KT/V, (*n* = 30)		1.47 (1.4–1.53)	1.15 (0.99–1.3)	1.49 (1.28–1.58)	1.3 (1.12–1.63)	1.12 (0.99–1.21)

* Successful completion of hemodialysis (HD) or hemodiafiltration (HDF) without significant problems with blood clotting. ⁑ Grade 1: no detectable clotting. Grade 2: minimal clot formation (fibrinous ring). grade 3: clot formation (up to 5 cm) but hemodialysis still possible; grade 4: complete occlusion of air traps or dialyzer rendering hemodialysis termination.

## Data Availability

All data generated or analyzed during this study are included in this published article.

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
