# Peer review of "Hemodialysis without Systemic Anticoagulation: A Randomized Controlled Trial to Evaluate Five Strategies in Patients at a High Risk of Bleeding"

_medsci, 2024, doi:10.3390/medsci12030038_

Round 1

Reviewer 1 Report (Previous Reviewer 1)

Comments and Suggestions for Authors

I thank you for your answers , even though you did not introduce  significant   changes in the manuscript related to my remarks.

Author Response

We appreciate the reviewer's input highlighting the limitations of our study, which we have incorporated into our discussion. While some of the points raised could not be addressed in this study, they will be considered in the design of future trials.

Reviewer 2 Report (Previous Reviewer 5)

Comments and Suggestions for Authors

The authors replied to all the questions I raised and I appreciate the last manuscript revised well.
I think this paper is suitable to be published in Medical Sciences.

Author Response

We appreciate the reviewer's comments and feedback. 

Reviewer 3 Report (New Reviewer)

Comments and Suggestions for Authors

Review for Medical Sciences on the article:

Hemodialysis without Systemic Anticoagulation: A Randomized Controlled Trial to Evaluate Five Strategies in Patients at High Risk of Bleeding

Pedro H F Gois, David McIntyre, Sharad Ratanjee, Anita Pelecanos, Carla Scuderi, Chungun L, Janoschka, Kara Summers, Haibing Wu, Belinda Elford, Dwarakanathan Ranganathan, Helen G Healy

The authors performed an open label randomized controlled trial on hemodialysis patients with contraindication to systemic heparinization. The flow-chart of the study is well depicted, the methods are really clear, the population is quite small but really well described, with a lot of treatments and datas taking into account. The English language is perfectly suitable for publication.

The results are in line with the methods and the discussion is in accordance with the results.

According to the limits, the authors described quite well which are the limits of the study.

Overall the work was really interesting and point out interesting hints on hemodialysis without heparin.

I have just a few questions:

How was the degree of inflammation of the considered patients?

Which were the values of ferritin and C-reactive protein?

Did any of the patients have a bacterial or a SARS-COV2 infection?

Were there differences in the dosage of erytropoiheting among groups?

Author Response

We appreciate the reviewer's feedback. Please find the responses to your questions below:

How was the degree of inflammation of the considered patients?

Response: Serum CRP was part of the planned data collection for included participants.

Which were the values of ferritin and C-reactive protein?

Response: Ferritin was not part of the pre-specified data collection, however, we did not observe statistically significant differences in serum CRP across all five groups (table. 

Did any of the patients have a bacterial or a SARS-COV2 infection?

Response: Our recruitment was significantly impacted by the COVID-19 pandemic due to the shutdown of elective surgeries. However, during the recruitment phase, Queensland's strict lockdown and border closures ensured that none of our study participants acquired COVID-19.

Were there differences in the dosage of erythropoietin among groups?

Response: Thank you for your question. There were no significant differences in ESA doses across Groups 1, 3, 4 and 5 (please see below). Despite randomisation, participants in group 2 were not on ESA. 

Mean weekly doses (in mcg) of darbepoetin alfa were (data not published):

Group  1           36.79 (SD 18.75)                      

Group 3           38.13 (SD 25.44)

Group 4            46.25   (SD 17.02)

Group 5             34.38 (SD 18.93)

Please note that participants on epoetin alpha had their doses converted to darbepoetin alpha according to the manufacturer.

Reviewer 4 Report (New Reviewer)

Comments and Suggestions for Authors

This is highly relevant study because there is only a few clinical trials comparing alternative interventions to systemic heparinization in HD. The research design and methodology are clearly presented and the conclusions support the results. However, there are few questions that need to be clarified.

1. What is the theoretical basis for using predilution HDF as an alternative to systemic anticoagulation in hemodialysis?

2. What were the objectives of introducing a predilution HDF intervention group in the recent trial?

3. What are the implications of over 20% premature clotting associated with predilution HDF for patients with a high risk of bleeding?

4. How might the findings from this study influence clinical practices regarding the use of predilution HDF in hemodialysis patients?

Author Response

We appreciate the reviewer's questions. Please find our responses below:

What is the theoretical basis for using predilution HDF as an alternative to systemic anticoagulation in hemodialysis?

Response: The rationale for using predilution HDF as an alternative to systemic anticoagulation in HD is based on the theory that infused dialysate would continuously rinse the dialyzer, thereby preventing the adherence of cells and proteins implicated in blood clotting.

Hemodial Int 2019, 23(4):426-432.

Artif Organs 2006, 30(2):106-110.

What were the objectives of introducing a predilution HDF intervention group in the recent trial?

Response: This was a result of multidisciplinary feedback provided by renal nurses, nurse practitioners, trainees, and nephrologists during our protocol design. Many clinicians in our unit were using HDF with predilution as a strategy for heparin-free HD with anecdotal observations of poor performances. Additionally, we aimed to test whether a second strategy for prevention of clotting (on top of predilution HDF) would provide an additive effect in preventing clotting of the extracorporeal circuit.

What are the implications of over 20% premature clotting associated with predilution HDF for patients with a high risk of bleeding?

Response: The implications include increased HD downtime, nursing staff workload, cost of the HD session, hospital waste generation, and ultimately patients' blood losses.

How might the findings from this study influence clinical practices regarding the use of predilution HDF in hemodialysis patients?

Response: After submitting our preliminary results to internal review, clinicians in our institution were more skeptical/cautious of using HDF with predilution alone for heparin-free HD.

This manuscript is a resubmission of an earlier submission. The following is a list of the peer review reports and author responses from that submission.

Round 1

Reviewer 1 Report

Comments and Suggestions for Authors

An interesting, well written paper. My observations mainly concern the study design. The most important shortcoming of this study is the absence of evaluation of dialysis with citrate-enriched dialysate, which is now possible with machines from all the major suppliers. We note that this was not feasible for the authors, but this remains a major limitation and makes the work a bit backwards.  

For the aim of this study, the ideal  would be for each patient to be treated with each of the procedures  described. This is obviously impossible if you want to do the study in the active presence of the risk of bleeding. It is therefore necessary that the different groups are perfectly matched

Although  the authors report only two differences (sex and catheters), the result of the analysis between the groups must be indicated in all the Tables in each of the variables indicated, together with the indication of the type of analysis performed.  

The number of patients and dialysis sessions per group is not entirely clear to me: 40 patients and 58 sessions as indicated at lines 195-197 ? This would be 1.5 session/patient!  What does N= 7, 4,8,7,6 (which in total makes 32) represent in Tables 1-3?   Patients or sessions?  I suppose sessions, but the title of Table 1 is Patient characteristics.  On the contrary, in Table 4, N= 11,8,17,11, makes 58  , I suppose sessions. Please clarify by writing the meaning next to the numbers.  

The list of the blood variables directly related to coagulation does not include some important elements, such as fibrinogen (with its very high molecular weight it plays a critical role on the coagulation risk during a 4-hour dialysis session). Furthermore, due to the specificity of this study, Antithrombin III  should also have been analyzed, above all to highlight any starting differences between the groups in their thrombotic risk. Finally, hematocrit is not provided in Table 1, it should be.

Reviewer 2 Report

Comments and Suggestions for Authors

Thank you for such work that deals with hot topics in nephrology (safe dialysis without anticoagulation).  Five small groups of ESKD patients were enrolled in this study with different options of dialysis without systemic anticoagulation.

I have a few comments for the authors:

-Some minor English typing mistakes will need to be revised.

-The authors mentioned that they enrolled 40 patients (8 patients in each group) but this was not the same in the tables. Please clarify this issue.

-Did the authors find any changes in the mean hemoglobin or mean KT/V in the group with a high clotting rate compared to other groups? 

Comments on the Quality of English Language

Please revise the English typing.

Reviewer 3 Report

Comments and Suggestions for Authors

In this study, the authors examined the effect of 5 different heparin-free HD treatments for up to 3 sessions on circuit clotting in totally 32 patients at high risk of bleeding. They found that success rates of HD/HDF sessions were higher in Group 2, 3 and 5 when compared to control group. This is an interesting study, while there are some concerns. 

1.     There was no statistical difference in primary and secondary outcomes among 5 groups. In addition, the numbers of participants were small. So, the impact of this study led to be weak.

2.     The reasons for heparin-free HD indications were quietly different among the 5 groups. In addition, there was no information for aspirin and direct oral anticoagulants prescription.

3.     Low molecular-weight heparin (LMWH) does not increase the risk of bleeding compared with unfractionated heparin (UFH) for the extracorporeal circuit anticoagulation in hemodialysis. Why the authors did not select LMWH in replacement of UFH?

Reviewer 4 Report

Comments and Suggestions for Authors

Authors compared the clotting outcome of five non-heparin dialysis methods. Their major limitation is the very small number of patients in each group which made their conclusion inconclusive. With such small number of patients, it is unlikely that the patients were in a basically similar state, especially in their vascular disease burden.  Another question is very tricky: why didn't they do their study in those patients without high risk of bleeding? There will be more patients and even less likelihood of unwanted complications.

Reviewer 5 Report

Comments and Suggestions for Authors

This study was conducted to evaluate the performance of combinations of interventions to prevent clotting of dialyzer/lines compared to single interventions in patients with high risk of bleeding who require heparin-free HD.

The authors showed that the combination of interventions, such as heparin-coated dialyzers combined with intermittent 0.9% saline flushes and predilution HDF combined with heparin-coated dialyzers, may have additive effects, leading to less frequent clotting and premature termination of a HD/HDF session. This is the first study evaluating the combination of saline flushes and predilution HDF with heparin-coated dialyzers to prevent clotting of the extracorporeal circuit in chronic HD patients with high risk of bleeding.

Although it is an interesting study, I have some comments.

1.    As successful completion of hemodialysis (HD) or hemodiafiltration (HDF) without significant problems with blood clotting is an important principle of blood purification therapy, it is reasonable to conclude that only heparin-coated dialyzers combined with intermittent saline flushes can be used as heparin-free HD.

2.    There was no difference in success rates between Group 1 (intermittent 0.9% saline flushes) and Group 4 (HDF with online predilution). What are the reasons for the different results in Group 3 (combination of heparin-coated dialyzer and intermittent 0.9% saline flushes) and Group 5 (combination of heparin-coated dialyzer and HDF with online predilution) ?